# Integrative Cross-Talk in Asthma: Unraveling the Complex Interactions Between Eosinophils, Immune, and Structural Cells in the Airway Microenvironment

**DOI:** 10.3390/diagnostics14212448

**Published:** 2024-10-31

**Authors:** Andrius Januskevicius, Egle Vasyle, Airidas Rimkunas, Kestutis Malakauskas

**Affiliations:** 1Laboratory of Pulmonology, Department of Pulmonology, Lithuanian University of Health Sciences, LT-44307 Kaunas, Lithuania; egle.jurkeviciute@lsmu.lt (E.V.); airidas.rimkunas@lsmu.lt (A.R.); kestutis.malakauskas@lsmu.lt (K.M.); 2Department of Pulmonology, Lithuanian University of Health Sciences, LT-44307 Kaunas, Lithuania

**Keywords:** eosinophil, asthma, cell–cell interaction, structural cells, immune cells

## Abstract

Asthma is a chronic inflammatory process that leads to airway narrowing, causing breath loss followed by spasms, wheezing, and shortness of breath. Within the asthmatic lungs, interaction among various immune cells and structural cells plays a significant role in orchestrating an inflammatory response in which eosinophils hold central importance. In these settings, allergens or other environmental exposures commonly drive the immune response to recruit eosinophils to the airways. The appearance of eosinophils in the airways indicates a dynamic interplay of various cell types within lung tissue and does not represent a passive effect of inflammation. The cellular cross-talk causes the persistence of eosinophilic inflammation, and if left untreated, it results in long-term damage to the airway structure and function. Further exacerbation of the condition occurs because of this. We discuss how this complex interplay of eosinophils, immune, and structural cells within the airway microenvironment leads to the distinct pathophysiological features in asthma, the variability in disease severity, and the response to biological treatments.

## 1. Introduction

Asthma is a chronic inflammatory disease of the airways; more than 300 million people worldwide are affected, and its prevalence continues to rise. The disease is characterized by airway hyperresponsiveness, obstruction of airflow, and chronic inflammation, leading to structural changes in the lung [1]. Among the several phenotypes of asthma, eosinophilic asthma is typified by increased levels of eosinophils in the airways. It represents one of the most severe and therapy-resistant manifestations of the disease [2]. Eosinophils play a critical role in promoting airway inflammation and remodeling, hence central to the pathogenesis of asthma [3].

The cellular interactions in asthma between eosinophils and other immune and structural cells have significant implications for further treatment approaches. In addition to maintaining inflammation, eosinophils communicate with structural cells, such as smooth muscle cells of the airways, fibroblasts, epithelial cells, and nerve cells. These interactions result in the release of pro-inflammatory mediators, the remodeling of the extracellular matrix (ECM), and the lesion of the epithelium, furthering disease progression and exacerbations. Understanding these complex interactions is necessary to explore new perspectives in managing asthma [4].

The purpose of this review is to gain insight into the role of the eosinophil in asthma and how interactions with immune and structural cells build up the progress of the disease. This review focuses on the interaction between eosinophils, bronchial epithelial cells, airway smooth muscle (ASM) cells, fibroblasts, and nerve cells and on how such interactions promote inflammation, tissue remodeling, and airway hyperresponsiveness. This review aims to summarize the cross-talk of eosinophils with different airway microenvironment cell types, including immune cells and structural cells. Further, the role of eosinophil-derived mediators, including cytokines, eosinophil extracellular traps (EETs), and exosomes, in promoting airway inflammation, epithelial damage, and tissue remodeling in asthma will be discussed. This review will cover the implications of such interactions for therapeutic approaches targeting eosinophils in asthma.

## 2. Eosinophils in Asthma, Surface Molecules and Receptors

Eosinophils are myeloid cells originating in the bone marrow and migrating to tissues, including the lungs, where they become activated during asthma. Eosinophils play a critical role in the pathogenesis of asthma, particularly in its more severe forms, often termed eosinophilic asthma. These white blood cells contribute to airway inflammation, which leads to airway hyperresponsiveness and remodeling.

Recent research has highlighted eosinophils as key players in asthma exacerbations, with their involvement in immune response and tissue damage. Eosinophils release mediators like major basic proteins, cytokines such as interleukin (IL)-5 and granulocyte-macrophage colony-stimulating factor (GM-CSF), and lipid mediators like cysteinyl leukotrienes [2,5]. These factors are critical in promoting inflammation and perpetuating asthma symptoms. Anti-IL-5 therapies have shown promise by reducing eosinophil levels and decreasing the frequency of asthma exacerbations in patients with severe eosinophilic asthma [6]. However, eosinophilic inflammation can persist even in the absence of IL-5 due to other immune pathways, particularly those involving Th2 cytokines (such as IL-4 and IL-13), which drive the recruitment and activation of eosinophils [7].

Furthermore, environmental triggers like viral infections, particularly those caused by rhinovirus, exacerbate eosinophilic activity by upregulating adhesion molecules such as intercellular adhesion molecule-1 (ICAM-1) on epithelial cells [8]. The investigation of eosinophilopoetin receptor gene expression in eosinophils revealed that in severe eosinophilic asthma, the gene expressions of IL-3Rα and IL-5Rα were significantly elevated compared with healthy controls, while the expression of GM-CSFRα showed no significant difference. In the allergic asthma group, IL-3Rα gene expression was moderately increased relative to healthy subjects, but IL-5Rα and GM-CSFRα were notably reduced [9].

The communication of eosinophils with the surrounding environment via surface proteins’ expression changes their activation state, including non-activated, pre-activated, ‘primed’, or fully activated [10]. Eosinophils express various surface markers that are important in their activation and function, especially in asthma. CD69, an early activation marker, is upregulated in eosinophils upon exposure to cytokines like IL-5, IL-3, and GM-CSF.

CD69 expression is low or absent in resting eosinophils but increases significantly in stimulated or inflamed states, such as asthma [11]. Moreover, increased levels of IL-10 were observed in CD69+ eosinophils. When CD69 was cross-linked with these cells, it significantly enhanced IL-10 production through the Erk1/2 pathway and promoted cell death via the JNK signaling pathway. These results suggest that the expression of CD69 in eosinophils acts as a functional immunosuppressive factor in type 2 inflammation [12].

L-selectin is typically expressed in circulating eosinophils and is downregulated in response to activation. It plays a role in eosinophil adhesion to endothelial cells, and its reduction is noted in activated eosinophils found in the bronchoalveolar lavage (BAL) fluid or sputum of asthma patients [13,14].

ICAM-1 is an adhesion molecule that plays a crucial role in mediating cell–cell interactions in inflammatory reactions. It is upregulated in eosinophils by various cytokines (IL-5, GM-CSF, IFN-γ), facilitating their binding to endothelial cells. It is increased in the airways of asthma patients but not significantly elevated in circulating eosinophils [15]. Moreover, an increased expression of ICAM-1 enhances eosinophil adhesion to the bronchial epithelium. This adhesion activates eosinophils, prompting them to release inflammatory mediators such as eosinophil cationic protein, contributing to airway inflammation and remodeling in asthma. In this context, eosinophils adhering to ICAM-1 can exacerbate asthma symptoms. ICAM-1 is involved in eosinophil transmigration through the vascular endothelium, leading to their accumulation in asthmatic airways, a key feature of chronic and severe asthma [8,16,17] (Figure 1).

CD44 functions as an adhesion molecule and hyaluronan receptor, mediating cell migration. Its levels increase in response to IL-5 and following antigen exposure [18,19]. In asthma, eosinophils in the airways exhibit higher CD44 expression, potentially contributing to tissue infiltration and inflammation. Anti-CD44 antibodies abolish eosinophil and lymphocyte infiltration into the airways and reduce Th2 cytokine, IL-4, and IL-5 levels [7].

P-selectin glycoprotein ligand-1 (PSGL-1) interacts with P-selectin on endothelial cells and platelets to mediate eosinophil rolling and adhesion. This interaction is particularly critical during Th2-driven inflammation where cytokines like IL-13 enhance P-selectin expression in endothelial cells, further promoting eosinophil adhesion and tissue infiltration [20]. It plays a significant role in eosinophil recruitment during the inflammatory process in asthma. Factors like IL-5 modulate its expression, and although its shedding occurs in vitro, it remains stable in vivo, especially in BAL eosinophils [21]. In the context of asthma, PSGL-1 and P-selectin interactions contribute to the recruitment of eosinophils into the airways, exacerbating asthma symptoms by increasing inflammation, mucus production, and airway hyperreactivity. Targeting this PSGL-1/P-selectin pathway has been proposed as a therapeutic strategy to reduce eosinophilic inflammation and mitigate asthma severity [22].

Integrins are adhesion receptors critical for eosinophil attachment to the endothelium and tissue migration. Eosinophils express several integrins, crucial for their role in adhesion, migration, and signaling during immune responses, especially in diseases like asthma. Integrins are heterodimeric proteins made up of α and β subunits, and eosinophils express a unique set of integrins: α4β1 (CD49d/CD29), αLβ2 (CD11a/CD18), α6β1 (CD49f/CD29), αXβ2 (CD11c/CD18), αMβ2 (CD11b/CD18), αDβ2, and α4β7 [10]. Integrin activation is tightly regulated, and different conformations (inactive, intermediate, high-activity) are associated with eosinophil activation states [23]. For example, αMβ2 (CD11b/CD18) is upregulated by IL-5 and other mediators, while intermediate and high-activity states of β1 and β2 integrins correlate with asthma severity and eosinophil activation [24]. Blocking eosinophil integrins can reduce their effect on pulmonary structural cells [25] (Figure 1).

Eosinophils express several cytokine receptors crucial for their activation, survival, and recruitment in asthma, particularly within the inflamed airways. These cytokine receptors regulate eosinophil functions in response to various signals and are, therefore, essential to the development of asthma. IL-5 receptor (IL-5R) is vital for eosinophil survival and activation, making it a key target in therapies like mepolizumab. IL-5 downregulates its receptor in eosinophils, leading to decreased IL-5 responsiveness. In contrast, IL-3R expression increases when eosinophils are exposed to IL-5, GM-CSF, or IL-3. This upregulation enhances eosinophil responsiveness to IL-3 and ensures their survival and activation in inflamed tissues. GM-CSFR is specifically downregulated by GM-CSF but not by IL-3 or IL-5, indicating distinct regulatory pathways for GM-CSF signaling. These findings highlight the complex regulation of eosinophil cytokine receptors, which can affect their activation and survival in asthma, influencing the effectiveness of therapies targeting these cytokines [9,26]. IL-13 and IL-4 receptors further contribute by promoting eosinophil recruitment and airway remodeling, with therapies like dupilumab targeting IL-4R [27].

Additionally, the thymic stromal lymphopoietin (TSLP) receptor plays a role in eosinophil recruitment, particularly in allergic asthma. Eosinophils express TSLP receptors (TSLPR), which are upregulated by pro-inflammatory cytokines like IL-3 and TNFα. This increased expression enhances eosinophils’ sensitivity to TSLP stimulation. TSLP stimulates eosinophil degranulation, leading to the release of eosinophil-derived neurotoxin (EDN). It also activates intracellular signaling, such as a signal transducer and activator of transcription 5 (STAT5) phosphorylation, which promotes eosinophil survival [28].

Fc receptors in eosinophils are involved in immune complex handling and cell activation. Eosinophils express Fc receptors for multiple immunoglobulins (Ig), including IgA, IgD, IgE, IgG, and IgM, allowing them to engage with the adaptive immune system [4]. The high-affinity FcεR1 receptor binds to IgE and signals through intracellular tyrosine kinases. While on mast cells and basophils, high-affinity IgE receptor (FcεR)1 is expressed as a tetramer (αβγ2), leading to degranulation upon IgE binding, and its expression on eosinophils is limited. Typically, eosinophils express FcεR1 as a trimer (lacking the β chain), and it does not contribute to their activation [29]. However, activation of eosinophils occurs through the cross-linking of FcαRI (IgA) and FcγRII (IgG), which has been shown to trigger eosinophil degranulation and other effector functions [30] (Figure 1).

## 3. Immune Cells in the Airway Microenvironment and Their Relations with Eosinophils

Asthma is a disease known to involve T helper (T_H_) cells. For a long time, it was described as an inflammatory disease with predominant T_H_ type 2 inflammation; however, now it is known that asthma might rise in T_H_17 manner with predominant neutrophilic inflammation. Type 2 or T_H_2 asthma is closely related to eosinophilic airway inflammation. At the early stage of studies, the findings of eosinophils in (BAL) fluid, induced sputum, or bronchial biopsies were considered the disease’s nature. The conception of eosinophilic inflammation is not related to the disease’s origin—it is predominant even in extrinsic or intrinsic asthma. However, it raises the question that the disease’s distinct trigger might be related to different pathophysiological mechanisms (Figure 2).

### 3.1. Th2 Cells

Th2 cells play a key role in asthma pathogenesis by secreting cytokines such as IL-4, IL-5, and IL-13, crucial for promoting allergic inflammation. Eosinophils, activated by Th2 cytokines, are a major factor in asthma severity. There is a direct correlation between eosinophil accumulation in the airways and disease severity or exacerbation frequency [31,32]. IL-5, initially discovered as a T cell-derived factor that activates B cells, is now recognized as crucial for eosinophil-dependent inflammation. It regulates eosinophil development, recruitment, and survival and is essential in developing bronchial hyperresponsiveness in Th2-mediated asthma [4,33,34]. The communication between eosinophils and Th2 cells is central to initiating and perpetuating allergic airway inflammation in asthma. Their interactions create a self-reinforcing cycle that leads to chronic inflammation, airway hyperresponsiveness, and structural remodeling of the airways. Th2 cells–eosinophils communication might be described as an amplification loop in asthma pathogenesis. Th2 cells release IL-5, promoting eosinophil growth and survival, while eosinophils release IL-4 and IL-13, reinforcing Th2 differentiation and function; moreover, eosinophils act as antigen-presenting cells (APCs), sustaining Th2 cell activation. After activation, Th2 cells and eosinophils produce chemokines, recruiting more eosinophils, Th2 cells, and other inflammatory cells to the airways. Eosinophils have been demonstrated to act as APCs in various experimental models of allergy [35]. They present antigens to primed T cells, enhancing Th2 cytokine production [36,37]. Upon migration to local lymph nodes, eosinophils localize within T cell-rich paracortical zones, promoting the expansion of CD4+ T cells. This activity is supported by increased expression of key surface molecules, such as CD80 and CD86, and major histocompatibility complex (MHC) class II markers of activated eosinophils [36,38]. Additionally, antigen-loaded eosinophils have been shown to stimulate the production of IL-5 when co-cultured with antigen-specific CD4+ T cells from allergic mice [36].

### 3.2. Dendritic Cells

Dendritic cells (DCs) are the primary antigen-presenting cells of the immune system, playing a crucial role in initiating and coordinating innate and adaptive immune responses. They specialize in recognizing and capturing foreign antigens at mucosal sites, then migrating to secondary lymphoid tissues to activate naive CD4+ T cells [39,40]. Depending on the DC subset, the local cytokine environment, and the activation signals received, DCs prime CD4+ T-helper (Th) cells, leading to their differentiation into various subsets, including Th1, Th2, Th9, and Th17 [41]. Eosinophils play a significant role in directing CD4+ T cell differentiation toward a Th2 phenotype by releasing preformed Th2-initiating cytokines, such as IL-4, IL-13, IL-6, and IL-25, upon stimulation. In addition to these cytokines, eosinophils release alarmins, which act as danger signals to activate local dendritic cells (DCs) [42,43].

Furthermore, eosinophils function as competent APCs, directly activating naive or previously primed CD4+ T cells. While the extent of eosinophils’ role in initiating Th2 responses to specific infectious agents or allergens is not fully understood, their involvement is crucial for generating protective Th2 immunity, as demonstrated in responses to S. stercoralis [42,43]. Additionally, eosinophil reconstitution has been shown to restore Th2 immunity in models of aerosolized ovalbumin challenge, even when introduced after sensitization but before the challenge [44]. An important area of ongoing research focuses on understanding the conditions, both infectious and genetic, where eosinophils and other innate immune cells may supplement the functions of professional APCs, such as DCs. Interestingly, eosinophils are not solely restricted to driving Th2 immunity. Their ability to present viral antigens and their possession of preformed Th1-inducing cytokines suggest that eosinophils can also promote Th1 immunity under specific conditions [44].

### 3.3. ILC2

ILC2s (type 2 innate lymphoid cells) play a significant role in asthma, with their numbers found to be elevated in the blood and BAL fluid of asthmatic patients [45]. In response to allergens, the expression of TNF-like cytokine (TL)1A, a ligand for death receptor (DR)3, is increased, contributing to the expansion and activation of ILC2s in humans [46]. In murine models, ILC2s have been shown to contribute to allergic lung inflammation by producing IL-5 and IL-13, promoting eosinophilia and bronchial hyperreactivity, particularly after exposure to house dust mite allergens. This process is enhanced in early life through IL-33 signaling [47,48,49].

Evaluating the direct effect between eosinophils and ILC2, William and colleagues highlight that depleting eosinophils led to significant reductions in total lung ILC2s, as well as IL-5+ and IL-13+ ILC2 subsets, across various respiratory inflammation models. This reduction was linked to decreased IL-13 levels and airway mucus production. Eosinophil-derived IL-4 and IL-13 were crucial for accumulating eosinophils and ILC2s in allergen-driven lung models. In vitro studies showed that eosinophils released soluble factors that promoted ILC2 proliferation and G protein-coupled receptor-mediated chemotaxis of ILC2s. When ILC2s were co-cultured with IL-33-activated eosinophils, both cell types exhibited changes in gene expression, indicating potential reciprocal interactions between eosinophils and ILC2s [50].

### 3.4. Mast Cells

Eosinophils and mast cells enhance each other’s activity in allergic disorders like asthma. Mast cells secrete IL-5, supporting eosinophil survival and activation, while eosinophil-derived major basic protein (MBP) activates mast cells, triggering the release of allergic mediators such as histamine and TNF-α [51]. Eosinophils respond to mast cell-derived prostaglandin D2 (PGD2) via the CRTH2 receptor, leading to chemotaxis and degranulation [52,53]. Studies show eosinophils and mast cells are physically connected in allergic disorders like EoE and mutually activate each other [54]. Anti-IL-5 therapies reduce eosinophil and mast cell numbers, highlighting eosinophils’ role in promoting mastocytosis in allergic diseases. The interaction between these cells is crucial for understanding allergic disease mechanisms and developing treatments.

### 3.5. B Cells

Once Th2 cells are formed in lung-draining lymph nodes, some interact with B cells, prompting their maturation into plasma cells that produce antibodies. Th2 cytokines, particularly IL-4 and IL-13, drive B cells to produce IgE. Although this process typically occurs in secondary lymphoid tissues, studies show that local IgE production can also happen in the lung mucosa of asthma patients [55]. IgE plays a crucial role in allergic asthma by binding to FcεRI receptors, found on basophils, mast cells, eosinophils, and dendritic cells, as well as on ASM, endothelial, and epithelial cells [56]. In asthma, memory IgG-positive B cells class switches to IgE under the influence of IL-4 and IL-13 from T follicular helper cells, becoming long-lived plasma cells [57]. A significant interaction between eosinophils and B cells has been recognized, where eosinophils process antigens, stimulate T cells, and contribute to humoral immune responses [58].

## 4. Eosinophils Interactions with Structural Cells

### 4.1. Eosinophil–Endothelial Cell Interaction

In asthma, eosinophils play a crucial role in the inflammation and remodeling of the airways, especially in severe eosinophilic asthma. Their interactions with endothelial cells are vital to their recruitment to inflamed airways. Eosinophils are recruited from the bloodstream to lung tissues by cytokines such as IL-5, which enhances their maturation and survival, and by chemokines like eotaxin (CCL11), which facilitates their migration across the endothelium [20].

Eosinophils interact with endothelial cells through a multistep process involving capture, rolling, adhesion, and transmigration. Initially, eosinophils roll along the endothelium, a process mediated predominantly by selectins (L-selectin, P-selectin) and integrins, such as α4β1 and α4β7 [59,60]. After rolling, eosinophils adhere firmly to the endothelium via vascular cell adhesion molecule-1 (VCAM-1), guided by chemokines and other signals, which allows them to transmigrate across the endothelial barrier [61].

This migration depends on the interaction between adhesion molecules, such as VCAM-1 on endothelial cells and integrins on eosinophils [62,63,64]. Cytokines like IL-4 and IL-13 promote the expression of VCAM-1, enhancing eosinophil adhesion and migration [65]. Moreover, anti-VCAM-1 antibody attenuates eosinophil recruitment in an OVA-induced murine allergic asthma model [66]. Once in the lung tissues, eosinophils release cytotoxic granules containing proteins like eosinophil cationic protein (ECP) and MBP, contributing to airway inflammation, damage, and hyperresponsiveness. The endothelial cells, when activated by inflammatory cytokines (e.g., TNF-α, IL-1β), further amplify eosinophil recruitment by increasing the production of chemokines like eotaxin and RANTES (CCL5) [67]. These interactions are key in the pathogenesis of asthma where eosinophils exacerbate inflammation and tissue remodeling, contributing to symptoms like bronchoconstriction and mucus hypersecretion.

GM-CSF enhances eosinophils’ survival, activation, and recruitment, particularly in allergic airway inflammation [68]. However, its effect on eosinophil rolling and adhesion to the endothelium is variable. In some cases, GM-CSF promotes the shedding of L-selectin, which reduces eosinophil rolling, while in others, it has little to no effect on L-selectin shedding [69]. This variability in response may be due to differences in GM-CSF receptor expression or individual donor variations.

The chemokines eotaxin, eotaxin-2, monocyte chemoattractant protein-4, and RANTES were found to induce eosinophil transendothelial migration in a concentration-dependent manner, with eotaxin and eotaxin-2 showing the highest potency. These chemokines facilitate eosinophil migration through interaction with the CCR3 receptor, as demonstrated by the complete inhibition of transendothelial migration when eosinophils were pretreated with anti-CCR3 antibodies. Additionally, activating endothelial cells with inflammatory cytokines IL-1β or TNF-α enhanced eosinophil migration, particularly when combined with eotaxin and RANTES. The study also highlighted the synergy between IL-5 and these chemokines in promoting eosinophil migration, while IL-5 did not show synergy with endothelial activators like IL-1β. Overall, the findings suggest that chemokines, particularly eotaxin and eotaxin-2, promote eosinophil migration in allergic inflammation via CCR3, and this process is amplified by cytokines like IL-1β, TNF-α, and IL-5 [70].

### 4.2. Eosinophil and Pulmonary Fibroblast Communication

In asthma, a complex interaction exists between pulmonary fibroblasts and immune cells, particularly eosinophils, contributing significantly to inflammation and fibrosis. Fibroblasts, traditionally considered structural cells responsible for ECM production, are now recognized for their active role in immune responses during asthmatic inflammation. These cells communicate with immune cells, such as eosinophils, mast cells, and T cells, influencing chronic inflammatory processes and airway remodeling [71].

Eosinophils, prominent in allergic asthma, release pro-inflammatory mediators like MBP and ECP during degranulation, which trigger fibroblast activation. Eosinophil-derived cytokines, such as IL-5 and IL-3, stimulate fibroblast production of key pro-inflammatory cytokines, such as IL-6, IL-8, and ICAM-1, which play a central role in neutrophil recruitment and inflammation [72]. Moreover, IL-1β plays a pivotal role in the eosinophil-derived soluble factors that drive pro-inflammatory responses in bronchial fibroblasts, which are linked to eosinophilic inflammation in the human airway following SBP-Ag exposure. Additionally, IL-1β-dependent NF-κB signaling collaborates with JAK/STAT and Src pathways to promote the release of IL-6 and IL-8 from human bronchial fibroblasts. This suggests that IL-1β is crucial for mediating paracrine signals from eosinophils undergoing cytolysis in allergic asthmatic airways [73].

Furthermore, eosinophils drive ECM protein production by fibroblasts, increasing collagen and fibronectin synthesis, contributing to the thickening and stiffening of airways, characteristic of asthmatic fibrosis. Esnaut S. and colleagues identified that the interaction between eosinophils and fibroblasts significantly alters gene expression in fibroblasts. RNA sequencing reveals that eosinophil degranulation products induce the upregulation of approximately 300 genes in lung fibroblasts, many of which are involved in immune response, tissue remodeling, and lipid metabolism. Key genes identified include C3, CXCL1, CXCL8, and IL-6, all promoting inflammation and further neutrophil activation. The upregulation of these genes underscores the role of fibroblasts in mediating eosinophil-driven inflammation [74]. Our studies highlight that eosinophils from asthmatic patients, particularly those with severe nonallergic eosinophilic asthma, significantly modified the expression of ECM proteins, matrix metalloproteinases, tissue inhibitors of metalloproteinases, and key components of both the canonical and noncanonical TGF-β signaling pathways in pulmonary fibroblasts [75]. Moreover, asthmatic eosinophils enhanced pulmonary fibroblast contractility and increased cell migration and the differentiation of these cells into a more contractile phenotype, enhancing contractile phenotype markers such as α-smooth muscle actin (α-sm-actin), smooth muscle myosin heavy chain (sm-MHC), sm-MLCK in ASM cells, and α-sm-actin [76].

The direct contact of eosinophil and pulmonary fibroblasts is important for their functions. A combined cell culture model between eosinophils and pulmonary fibroblasts highlights an increased adhesion of eosinophils in asthma and prolongs their survival after direct contact with fibroblasts, potentially enhancing their pro-proliferative effects [77].

The cross-talk between eosinophils and fibroblasts promotes fibroblast-to-myofibroblast differentiation, a process mediated by transforming growth factor-beta (TGF-β) released by eosinophils. Myofibroblasts enhance ECM deposition, increasing the levels of collagen types I and III, fibronectin, and elastin, which further drive airway remodeling and fibrosis in asthma [78]. This fibroblast activation and matrix remodeling play a crucial role in the irreversibility of airway obstruction in chronic asthma. This interaction is also linked to the release of matrix metalloproteinases (MMPs) by fibroblasts, which degrade and remodel the ECM, contributing to abnormal matrix turnover. TGF-β and MMPs are crucial for regulating this dynamic as their imbalances lead to excessive ECM deposition, promoting airway fibrosis and structural changes.

### 4.3. Eosinophils—Airway Bronchial Smooth Muscle Cells

ASM primarily develops from mesenchymal precursors during embryogenesis, playing a key role in airway tree development [79]. Found throughout the respiratory tree, ASM helps maintain basal tone and modulate airflow resistance, though their precise physiological role post-development remains debated due to experimental challenges [80,81]. Despite this, ASM’s role in airway diseases, particularly asthma, is well established [82]. Airway remodeling includes structural changes such as ASM alterations, while airway hyper-responsiveness involves an exaggerated response to stimuli, primarily driven by bronchial smooth muscle activity, with inflammation playing a significant role in this process [83,84].

Banerjee and colleagues explored the regulatory relationships between ASM genes to uncover potential mechanisms in humans. Using RNA-Seq, gene expression was measured in both asthmatic and non-asthmatic groups. A gene network was constructed, prioritizing 121 differentially expressed genes and 116 transcription factors. The asthmatic group showed a loss of gene connectivity due to the rewiring of major regulators, with transcription factors like ZNF792, SMAD1, and SMAD7 being differentially correlated. These genes and transcription factors were linked to pathways involved in herpes simplex virus infection, Hippo and TGF-β signaling, adherents, gap junctions, and ferroptosis [85]. However, it results from the complex interactions between cells and molecules during chronic asthma inflammation; eosinophils are one of the most important cells, leading to alterations in ASM.

Direct contact with eosinophils can change ASM cells’ physiological activity. Eosinophils can adhere to ASM cells, promote the expression of TGF-β1, WNT-5a, and ECM protein fibronectin and collagen gene expression, and stimulate ASM cell proliferation in vitro. These effects were significantly stronger when ASM cells were co-cultured with eosinophils from asthmatic patients [86]. Halwani et al. found that increased ASM proliferation after co-culturing with eosinophils depended on the release of CysLTs by eosinophils. This release required direct cell-to-cell contact between eosinophils and ASM cells. These findings suggest that eosinophils play a role in airway remodeling in asthma by promoting ASM proliferation, thereby increasing ASM mass [87]. Moreover, direct contact with ASM cells prolongs eosinophils viability [77], and the lung-resident eosinophils subtype has a more pronounced effect on ASM cell proliferation and viability than inflammatory eosinophils [88]. Furthermore, asthmatic eosinophils enhanced collagen gel contraction by ASM cells, increased their migration, promoted their differentiation into a more contractile phenotype [76], and promoted ECM components production [75], mainly via the TGF-β1 signaling pathway in asthma [89].

Eosinophils-released exosomes also have a direct effect on ASM cells. These exosomes alter various processes within these cells and change the expression of several pro-inflammatory molecules, contributing to asthma pathogenesis [90]. Specifically, eosinophil-derived exosomes increased the gene expression of CCR3 and VEGFA in ASM cells after 24 h. CCR3 plays a key role in triggering cell proliferation via the MAPK pathway, and its activation likely leads to the abnormal proliferation of ASM cells, a hallmark of airway remodeling in asthma [91]. The increased expression of VEGFA, a factor associated with bronchial wall remodeling, further supports the contribution of eosinophil exosomes to these remodeling processes [91,92,93].

### 4.4. Eosinophil—Nerve Cells Interaction

Airway sensory nerves detect stretch and chemical stimuli and relay information to the central nervous system via the vagus nerve, which also carries parasympathetic fibers responsible for bronchoconstriction [94,95]. Parasympathetic nerves release acetylcholine (ACh), which stimulates M3 muscarinic receptors on ASM, causing contraction, while M2 receptors limit ACh release to regulate bronchoconstriction [96,97]. Sensory and parasympathetic nerves communicate through a neuronal reflex pathway that triggers “reflex bronchoconstriction” in response to stimuli like methacholine, histamine, cold air, exercise, and allergens, which are heightened in asthma patients [98,99].

Multiple studies suggest a relationship between eosinophils and neuronal homeostasis. For instance, peripheral dorsal root ganglia and airway neurons secrete eotaxins that recruit eosinophils to these regions [100,101]. Eosinophils can also produce nerve growth factors and neurotrophins like NT-3 at both mRNA and protein levels, which directly modulate neuronal activity, a process further amplified through Fc receptor-mediated eosinophil activation [102]. Both murine and human eosinophils have been observed to promote dorsal root ganglia neuron branching in vitro without the need for direct contact, potentially explaining nerve outgrowth and neurological symptoms seen in atopic dermatitis [100]. In asthma, lung neurotrophins prolong eosinophil survival, potentially leading to a self-perpetuating cycle that exacerbates the disease [103]. This connection may also help account for the neuronal hypersensitivity in allergic diseases, including asthma [104]. In a guinea pig asthma model, eosinophils migrate into nerves, particularly the vagal nerve, via a CCR3-dependent pathway [101] and release MBP, which acts as a muscarinic receptor 2 antagonist [105]. This action enhances acetylcholine release, worsening bronchoconstriction. In human asthma, eosinophils are often found near nerve endings, with extracellular MBP adhered to these regions, indicating a regulatory role for eosinophils in neuronal function [106].

In the newest studies, Drake and colleagues identified that airway epithelial sensory nerves undergo significant structural remodeling in eosinophilic asthma, characterized by increased nerve density in the airways. Additionally, airway eosinophils enhance sensory innervation of the epithelium and neuronally mediated airway responsiveness in a transgenic mouse model. These results establish a crucial connection between eosinophilic inflammation and airway nerve growth, suggesting that sensory neuroplasticity may play a role in exacerbating symptoms and worsening lung function in asthma [107]. Airway sensory afferents play a key role in regulating airway physiology and maintaining lung health, highlighting the potential for new therapeutic developments targeting sensory nerves.

### 4.5. Eosinophils Cross-Talk with Bronchial Epithelium

The bronchial epithelium exhibits structural and functional abnormalities in asthma, often due to chronic injury and defective repair mechanisms. These epithelial injuries, resulting from various insults such as allergens, pollutants, or pathogens, cause significant changes in the airway’s integrity, promoting immune cell recruitment, inflammation, and epithelial shedding [108,109]. As the first line of defense, the bronchial epithelium is equipped with various receptors capable of recognizing and responding to allergens, viruses, and other stimuli, and it serves as a barrier and critical interface between the external environment and the underlying tissues [110]. In individuals with asthma, the bronchial epithelium is characterized by increased sensitivity to various factors, resulting in an increased immune response and inflammation. When epithelial cells recognize a threat, they release inflammatory mediators, such as cytokines and chemokines. These signaling molecules play a role in the recruitment and activation of immune cells, mainly eosinophils, contributing to the inflammatory response [111].

The most common asthma phenotype is type 2 or eosinophilic asthma. Eosinophils, whose maturation in the bone marrow depends on IL-5, actively contribute to the innate and adaptive inflammatory cascades by producing and releasing various chemokines, cytokines, lipid mediators, and other growth factors [4]. The bronchial mucosa, close to the epithelial layer, is the preferred site of eosinophil accumulation in asthma where these cells are exposed to many substances produced by the epithelium, such as type 2 cytokines and alarmins. In turn, eosinophils are stimulated to produce and release several mediators that act on the epithelium and promote airway remodeling. Thus, cross-talk between eosinophils and bronchial epithelial cells may play a significant role in the pathogenesis of asthma [112].

During the initial allergen exposure, known as sensitization, allergens penetrate the epithelial barrier and are processed by dendritic cells, which present them to naïve T cells. That triggers B cells to produce IgE, which binds to Fcε receptors on mast cells, priming the immune system for future responses [113,114]. Upon subsequent allergen exposure, mast cells are activated, releasing histamine and inflammatory mediators like IL-8, IL-13, and TNF-α. These mediators increase vascular permeability, facilitating immune cell migration and stimulating goblet cells to produce more mucus [115]. Eosinophil recruitment during the late-phase allergic reaction leads to ongoing airway inflammation [116]. Moreover, the bronchial epithelium responds to irritants, such as allergens and pollutants, by releasing specific cytokines, including IL-25, IL-33, and TSLP. These cytokines, often called alarmins, initiate an inflammatory cascade that activates Th2 cells and ILC2s. This activation results in the enhanced production of cytokines like IL-4, IL-5, and IL-13, crucial for recruiting and activating eosinophils, thereby amplifying the inflammatory process [65,117].

Eosinophils play a central role in epithelial damage in asthma. Their infiltration and subsequent degranulation release cytotoxic proteins, such as eosinophil basic protein, damaging epithelial cells, increasing mucus production, and inducing airway constriction. Several studies have shown that the major eosinophil chemoattractant eotaxin-3 is strongly induced by IL-13 in airway epithelial cells [118,119]. Salter et al. show that bronchial epithelial cells release factors that promote eosinophilopoiesis, with the potential of increasing as asthma severity worsens. The epithelial-derived alarmin TSLP significantly promotes eosinophil-lineage commitment and differentiation, especially when combined with IL-5 in vitro, suggesting that bronchial epithelial cells contribute to persistent eosinophilic inflammation in asthma. The results support the hypothesis that bronchial epithelial cell-derived factors stimulate local eosinophil production from bone marrow-derived progenitor cells in asthmatics. This ongoing eosinophilopoiesis may perpetuate airway eosinophilia, contributing to tissue remodeling in chronic asthma [120]. Moreover, eosinophils-derived exosomes elevate the apoptosis rate in epithelial cells and slow the repair of existing epithelial damage. Additionally, exosomes from eosinophils of asthmatic patients upregulate the gene expression of TNF, CCL26, and POSTN, thereby extending the inflammatory state [91].

EETs disrupt airway integrity by causing airway epithelial detachment, leading to increased epithelial permeability and heightened inflammation in asthmatic airways. The detachment of epithelial cells results in cell death, as these cells rely on adhesion to the basement membrane for survival [121]. Eosinophils produce EETs that enhance the production of IL-8 in airway epithelial cells, which confirms their pro-inflammatory effects [122]. Moreover, targeting ECP significantly reduces EET-induced pro-inflammatory cytokine production in airway epithelial cells [122].

Eosinophils and bronchial epithelial cells engage in significant cross-talk in asthma. Eosinophils contribute to airway inflammation and remodeling by releasing various mediators that damage the epithelium and promote mucus production. Eosinophil-derived factors, including EETs and exosomes, further disrupt airway integrity by increasing epithelial permeability, slowing tissue repair, and extending the inflammatory state, particularly in severe asthma.

## 5. Conclusions

The present review discussed the multilayered role of the eosinophil in asthma and their interplay with diverse immune and structural cells within the airway microenvironment. Since eosinophils are at the frontline of the pathogenesis of eosinophilic asthma, they hold a very central position in driving airway inflammation and contributing to some characteristic structural changes, such as airway remodeling and hyperresponsiveness. Their interaction with the bronchial epithelium cells, ASM, fibroblast, and nerves is very important for perpetuating the disease cycle.

Key findings underline that eosinophils interact with structural cells in several ways by releasing cytokine, EETs, and exosomes, leading to disrupted epithelium integrity, proliferating ASM, and increasing nerve hyperresponsiveness. All these processes contribute to the development of perpetuating inflammation, airway remodeling, and deteriorating symptomatology in asthma. It also emphasized how eosinophil-derived mediators, including IL-5, TGF-β, and VEGFA, contribute to tissue damage and airway remodeling, thus perpetuating asthma. Realization of the complexity of such interactions identifies possible targets of therapy.

The identification of eosinophilic pathways and the IL-5 axis, integrin signaling, and exosome release offers new targets for developing novel treatments aimed at reducing airway inflammation and remodeling by improving outcomes in patients with severe asthma.

## Figures and Tables

**Figure 1 diagnostics-14-02448-f001:**
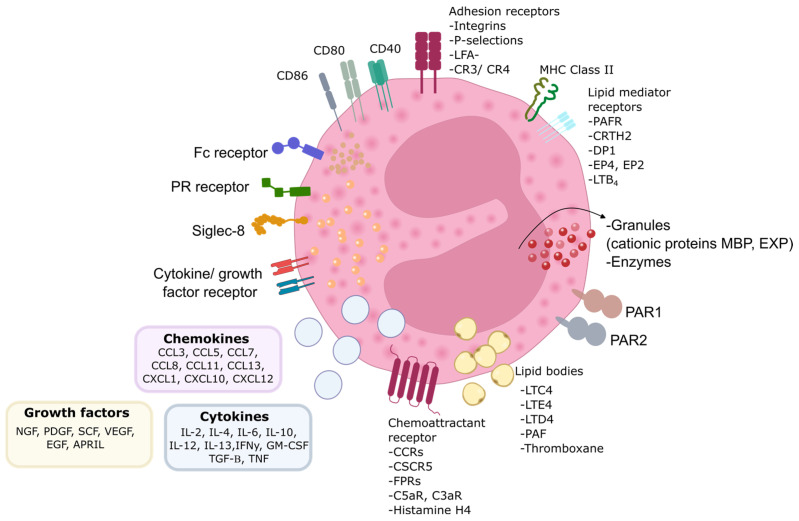
Eosinophil biology. Eosinophils are equipped with numerous proteins, receptors, and enzymes that enable them to interact with their surroundings and express various surface receptors for factors involved in cell growth, survival, adhesion, migration, and activation. Alongside these receptors, eosinophils express adhesion molecules on their surface, facilitating their migration and response to different stimuli. These interactions play a crucial role in their function within the immune system. PR—pattern recognition receptors; MBP—major basic protein; EPX—eosinophil peroxidase; MHC—major histocompatibility complex; CD—cluster of differentiation; CCR —CC-chemokine receptor; CXCR—CXC-chemokine receptor; FPR—formyl peptide receptor; C5aR—complement component 5a receptor; C3aR—complement component 3a receptor; LFA-1—lymphocyte function-associated antigen 1; CR—complement receptor; LTC4—leukotriene C4; LTE4/D4—leukotriene E4; leukotriene D4; PAF—platelet-activating factor; PAR—protease activated receptor; PAFR—platelet-activating factor receptor; CRTH2—chemoattractant receptorhomologous molecule expressed on T-helper type 2 cells; DP1—prostaglandin D2 receptor 1; EP4—prostaglandin E2 receptor 4; LTB4—leukotriene B4; GM-CSF—granulocyte-macrophage colony-stimulating factor; IL—interleukin; IFNγ—interferon γ; TGF-β—transforming growth factor β; TNF—tumor necrosis factor; NGF—nerve growth factor; PDGF —platelet-derived growth factor; SCF—stem cell factor; VEGF—vascular endothelial growth factor; EGF—epidermal growth factor; APRIL—proliferation-inducing ligand; CCL—C-C chemokine ligand; Fc—fragment crystallizable region; CXCL—C-X-C chemokine ligand.

**Figure 2 diagnostics-14-02448-f002:**
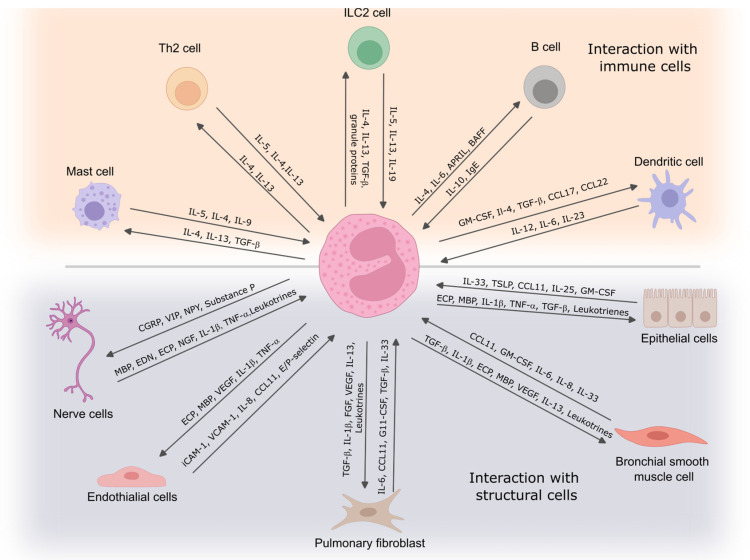
Eosinophil interaction with immune and structural cells. Eosinophils communicate with immune cells to regulate inflammation and contribute to immune responses, while their interactions with structural cells influence tissue remodeling and airway function. These cell-to-cell interactions play a critical role in the development and persistence of asthma symptoms. IL—interleukin; TGF-β—transforming growth factor β; Ig—immunoglobulin; APRIL—proliferation-inducing ligand; CCL—C-C chemokine ligand; TSLP—thymic stromal lymphopoietin; ECP—eosinophil cationic protein; MBP—eosinophil basic protein; TNF—tumor necrosis factor; VEGF—vascular endothelial growth factor; FGF—fibroblast growth factor; ICAM-1—intercellular adhesion molecule 1; VCAM-1—vascular cell adhesion molecule 1; NGF—nerve growth factor; BAFF—B cell-activating factor; CGRP—calcitonin gene-related peptide; VIP—vasoactive intestinal peptide; NPY—neuropeptide Y.

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
