# Peer review of "Integrative Cross-Talk in Asthma: Unraveling the Complex Interactions Between Eosinophils, Immune, and Structural Cells in the Airway Microenvironment"

_diagnostics, 2024, doi:10.3390/diagnostics14212448_

Round 1
Reviewer 1 Report
Comments and Suggestions for Authors
The peer-reviewed manuscript "Integrative Interactions in Asthma: Unraveling the Complex Interactions between Eosinophils, Immune, and Structural Cells in the Airway Microenvironment" is a review article focusing on the role of eosinophils in asthma and their interactions with immune and structural cells in the airway microenvironment. The authors focus on the complex interactions between eosinophils, bronchial epithelial cells, airway smooth muscle cells (ASMCs), fibroblasts, and neural cells. The manuscript provides an extensive review of the literature on eosinophil interactions with various cells in the context of asthma. The authors cover a wide range of aspects, including the mechanisms of eosinophil activation, their interaction with immune cells (T lymphocytes, dendritic cells, ILC2, mast cells, B cells), and their impact on structural cells (endothelium, fibroblasts, GMDP, nerve cells).
The authors describe in detail the mechanisms of interaction between eosinophils and various cell types, including the molecular mechanisms of adhesion, signaling pathways, and the cytokines and chemokines involved.
The authors take into account current data on the role of eosinophils in the development of asthma, including the involvement of eosinophil extracellular traps (EETs) and exosomes.
The manuscript highlights the clinical significance of the interaction between eosinophils and various cells, indicating the possibility of developing new therapeutic strategies targeting eosinophils.
Weaknesses:
The manuscript is mainly a review of existing data, but does not contain an in-depth critical analysis. The authors do not use a systematic approach to selecting literature.
Some sections of the manuscript could be better structured. For example, the section "Eosinophils and GMDP" could be divided into subsections reflecting different mechanisms of interaction. The manuscript ends with a brief conclusion on the significance of the interaction between eosinophils and various cells, but does not offer specific recommendations for further research or the development of new therapeutic strategies.
Recommendations:
The authors should include a critical appraisal of the existing data and express their own opinion on controversial issues. The authors can use a systematic approach to selecting literature to ensure the completeness and representativeness of the review.
The authors can improve the structure of the manuscript, including a clearer division of sections and subsections.
Overall:
The manuscript is a valuable review of the literature on the interaction of eosinophils with various cells in the context of asthma. Despite some shortcomings, the manuscript is a valuable resource for researchers interested in this direction. I believe that it can be both improved and accepted for publication in its current form.
Reviewer 2 Report
Comments and Suggestions for Authors
Dear authors and editor
It is a comprehensive review that highlighted the complex relationship between airways immune and structural cells in asthma, focusing on the role of eosinophils.
The article could be published in the present form